# Tunable Synthesis of Hollow Co$_3$O$_4$ Nanoboxes and Their Application in Supercapacitors

**Xiao Fan, Per Ohlckers** *[ORCID] **and Xuyuan Chen** *

Department of Microsystems, Faculty of Technology, Natural Sciences and Maritime Sciences, University of South-Eastern Norway, Campus Vestfold, Raveien 215, 3184 Borre, Norway; xiao.fan@usn.no

* Correspondence: per.ohlckers@usn.no (P.O.); xuyuan.chen@usn.no (X.C.); Tel.: +47-310-09-315 (P.O.); +47-310-09-028 (X.C.)

**Abstract:** Hollow Co$_3$O$_4$ nanoboxes constructed by numerous nanoparticles were prepared by using a facile method consisting of precipitation, solvothermal and annealing reactions. The desirable hollow structure as well as a highly porous morphology led to synergistically determined and enhanced supercapacitor performances. In particular, the hollow Co$_3$O$_4$ nanoboxes were comprehensively investigated to achieve further optimization by tuning the sizes of the nanoboxes, which were well controlled by initial precipitation reaction. The systematical electrochemical measurements show that the optimized Co$_3$O$_4$ electrode delivers large specific capacitances of 1832.7 and 1324.5 F/g at current densities of 1 and 20 A/g, and only 14.1% capacitance decay after 5000 cycles. The tunable synthesis paves a new pathway to get the utmost out of Co$_3$O$_4$ with a hollow architecture for supercapacitors application.

**Keywords:** Co$_3$O$_4$; hollow nanoboxes; supercapacitors; tunable synthesis

## 1. Introduction

Nowadays, the growing energy crisis has enormously accelerated the demand for efficient, safe and cost-effective energy storage devices [1,2]. Because of unparalleled advantages such as superior power density, strong temperature adaptation, a fast charge/discharge rate and an ultralong service life, supercapacitors have significantly covered the shortage of conventional physical capacitors and batteries, and sparked considerable interest in the last decade [3–5]. Supercapacitors are generally classified into electric double layer capacitors storing energy depending on reversible ions absorption at the interface [6–9] and pseudocapacitors through reversible faradaic redox reactions [10–13]. The different energy storage mechanisms directly determine whether electric double layer capacitors utilize carbonaceous materials as electrode materials [6,7,14], while pseudocapacitors employ transition metal oxides/hydroxides [10,11,15]. Drawing a benefit from faradaic redox reactions, pseudocapacitors usually generate several times higher specific capacitance than that of electric double layer capacitors, and have attracted major attention in recent years [15,16].

Among various transition compounds, Co$_3$O$_4$ is extensively recognized as one of the most representative pseudocapacitive materials owing to its abundant resources, environmental friendliness, low cost, strong redox activity and more importantly its superior theoretical specific capacitance (3560 F/g) [17–20]. The series of excellent features have triggered tremendous efforts on Co$_3$O$_4$ research for high performance supercapacitors application. To date, various solid or hollow structures of Co$_3$O$_4$ have been achieved. The previous progress illustrated the hollow architecture that a highly porous morphology could provide for a large electroactive surface area with extra active sites and facile ions transportation, consequently enhancing the electrochemical properties dramatically [21,22]. For instance, Zhou et al. reported hollow Co$_3$O$_4$ cages and presented the specific capacitance of 948.9 F/g at

1 A/g (536.8 F/g at 40 A/g) [23]. The hollow $Co_3O_4$ nanotubes prepared by Yao et al. demonstrated the specific capacitance of 1006 F/g at 1 A/g (512 F/g at 10 A/g) [24]. Zhu et al. synthesized hollow $Co_3O_4$ spheres and hollow $Co_3O_4$ microflowers. The spheres deliver the specific capacitance of 342.1 F/g at 0.5 A/g (235 F/g at 10 A/g), while the microflowers show the specific capacitance of 210 F/g at 0.5 A/g (180 F/g at 10 A/g) [25]. However, the obtained specific capacitances up to now are far lower than their maximum theoretical value. Moreover, the $Co_3O_4$ also suffers from a poor rate performance. Obviously, both low specific capacitance and poor rate performance severely hinder the practical application of $Co_3O_4$. Therefore, even though major contributions are achieved, there is still much room to deeply investigate the ability of the hollow $Co_3O_4$ to reach a large specific capacitance closer to its maximum theoretical value, as well as an excellent rate capability. Recently, metal-organic frameworks (MOFs), in which zeolitic imidazolate framework (ZIF) is most representative, have received much attention for constructing a hollow nanostructure for supercapacitor applications. By using the ZIF as a self-sacrificial template, a well-defined hollow structure as well as a highly electrolyte ions accessible surface area can be achieved [26–28]. Unfortunately, it is difficult to control the sizes of nanoarchitectures via a facile approach at present, which fails to further enhance the electrochemical performance.

In this paper, hollow $Co_3O_4$ nanoboxes were successfully prepared from ZIF-67 precursors. The hollow nanostructure and highly porous morphology of the $Co_3O_4$ synergistically enhance the electrochemical performances. Meanwhile, in this design, the sizes of the nanostructures were well controlled by the precipitation reaction. There are scarce reports about systematically tunable sizes of architectures in hollow $Co_3O_4$ preparation for high performance supercapacitors. What is more, this distinctive advantage makes the hollow $Co_3O_4$ nanoboxes comprehensively utilized. In this work, the optimized $Co_3O_4$ electrode exhibits an unprecedented specific capacitance (1832.7 F/g at 1 A/g), while the specific capacitance is still as high as 1324.5 F/g at 20 A/g. In addition, the capacitance only decays 14.1% after 5000 cycles. The pseudocapacitive performances mark the obtained $Co_3O_4$ as a promising electrode material candidate for supercapacitors.

## 2. Materials and Methods

### 2.1. Synthesis of Dodecahedral Diamond ZIF-67

Dodecahedral diamond ZIF-67 with different diameters were prepared by a precipitation method. To achieve this, 0.273, 0.102 or 0.068 M of 2-methylimidazole methanol solution was added to 0.04 M of cobalt nitrate methanol solution. The mixed solution was stirred for 30 min. Then the cloudy solution was kept static for 24 h at room temperature. The purple precipitates (denoted as ZIF-67-1, ZIF-67-2 and ZIF-67-3) were obtained and washed with methanol via centrifugation.

### 2.2. Conversion to Hollow Co-LDH (Co-Layered Double Hydroxide)

The hollow structures were transformed from ZIF-67 (-1, -2, -3) by using a solvothermal reaction. Afterwards, 0.05 g of ZIF-67 (-1, -2, -3) was dispersed in 0.01 M cobalt nitrate methanol solution and stirred for 10 min. After that, the pink slurry was transferred to a Teflon-lined stainless-steel autoclave and kept at 120 °C for 1 h. The brown precipitates (named as Co-LDH-1, Co-LDH-2 and Co-LDH-3) were obtained and washed with methanol via centrifugation.

### 2.3. Preparation of Hollow $Co_3O_4$ Nanoboxes

The Co-LDH (-1, -2, -3) were dried in an oven at 60 °C overnight and calcinated at 500 °C for 2 h. The final products were called $Co_3O_4$-1, $Co_3O_4$-2 and $Co_3O_4$-3.

### 2.4. Material Characterizations

The morphological and structural features of as-prepared samples were examined by scanning electron microscope (SEM, Hitachi SU8230) at 10 kV. The crystallinity and crystal phase were investigated

through X-ray powder diffraction (XRD, EQUINOX 1000) with Cu-K$\alpha_1$ radiation ($\lambda$ = 1.5406 Å) at a scanning speed of 6°/min.

### 2.5. Electrode Fabrication

Nickel foam (NF, 1.0 cm × 1.0 cm) was selected as the current collector. The as-prepared $Co_3O_4$ (80 wt%), acetylene black (15 wt%) and Polyvinylidene fluoride (PVDF, 5 wt%) were added to N-Methyl-2-pyrrolidone (NMP) and grinded in a mortar thoroughly. The resulting mixture was pressed on NF and dried in an oven at 100 °C overnight. The loading mass was approximately 1 mg.

### 2.6. Electrochemical Measurements

Cyclic voltammetry (CV) at typical scan rates (5, 10, 20, 50 and 100 mV/s), galvanostatic charge/discharge (GCD) at representative current densities (1, 2, 5, 10 and 20 A/g) and electrochemical impedance spectroscopy (EIS) in wide frequency range (100 mHz–100 kHz) were conducted on an electrochemical workstation (Zahner IM6). The configuration, potential window, electrolyte, counter electrode and reference electrode were set as three-electrode system, 0–0.65 V, 2 M KOH, Pt net and Ag/AgCl (3.5 M KCl), respectively.

The specific capacitance of the active material is calculated based on a GCD test according to Equation (1) [29]:

$$C = \frac{I\Delta t}{m\Delta V},\tag{1}$$

where: *C* (F/g) is the specific capacitance; *I* (A) is the discharge current; $\Delta t$ (s) is the discharge time; $\Delta V$ (V) is the voltage window; and *m* (g) is the mass of active material.

## 3. Results

Figure 1 presents the XRD patterns of the calcination products. The identified diffraction peaks of all samples can be assigned to the face centered $Co_3O_4$ (JCPDS No. 42-1467, space group: *Fd3m (227)*), suggesting a high purity of the samples [30]. According to the Scherrer equation [31], the crystallite sizes of $Co_3O_4$-1, $Co_3O_4$-2 and $Co_3O_4$-3 are around 23.6, 20.0 and 24.3 nm, respectively.

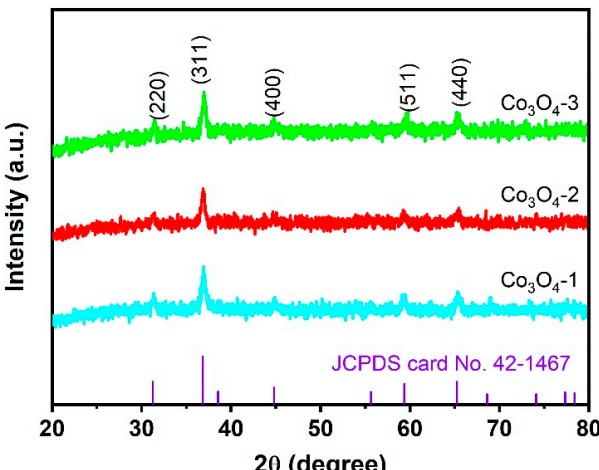

**Figure 1.** X-ray powder diffraction (XRD) patterns of $Co_3O_4$-1, $Co_3O_4$-2 and $Co_3O_4$-3.

Typical SEM images of ZIF-67 are shown in Figure 2a–c. The ZIF-67 present monodispersity and the average sizes of ZIF-67-1, ZIF-67-2 and ZIF-67-3 are 0.39 ± 0.05, 1.98 ± 0.18 and 3.28 ± 0.37 μm, respectively. Namely, the average sizes increase with the decrease of 2-methylimidazole concentration. This phenomenon is attributed to the stepwise formation of ZIF-67. The process can be briefly divided into a nucleation phase and growth stage. In the initial nucleation phase, the

increased concentration of 2-methylimidazole enhanced the nucleation rate, determining a small size of each unit cell. Subsequently, diffusive particles slowly grew around the cell to form ZIF-67 [32,33]. After the conversion, the ZIF-67 transferred to hollow Co-LDH with a significant morphology change. Figure 2d–f shows that there were nanosheets vertically growing on the surface of as-formed ZIF-67 and the average sizes shrunk a little bit. Meanwhile, with the continuous dissolution of the inner cobalt, the hollow Co-LDH nanoboxes with different sizes were formed. Hollow $Co_3O_4$ were obtained through calcination, as displayed in Figure 2g–i. Beyond expectations, the overall size and hollow feature were well maintained even after heat treatment, except for $Co_3O_4$-1 due to the very thin configuration of the shell. It could be dimly seen that the shell of hollow $Co_3O_4$ consisted of small nanoparticles.

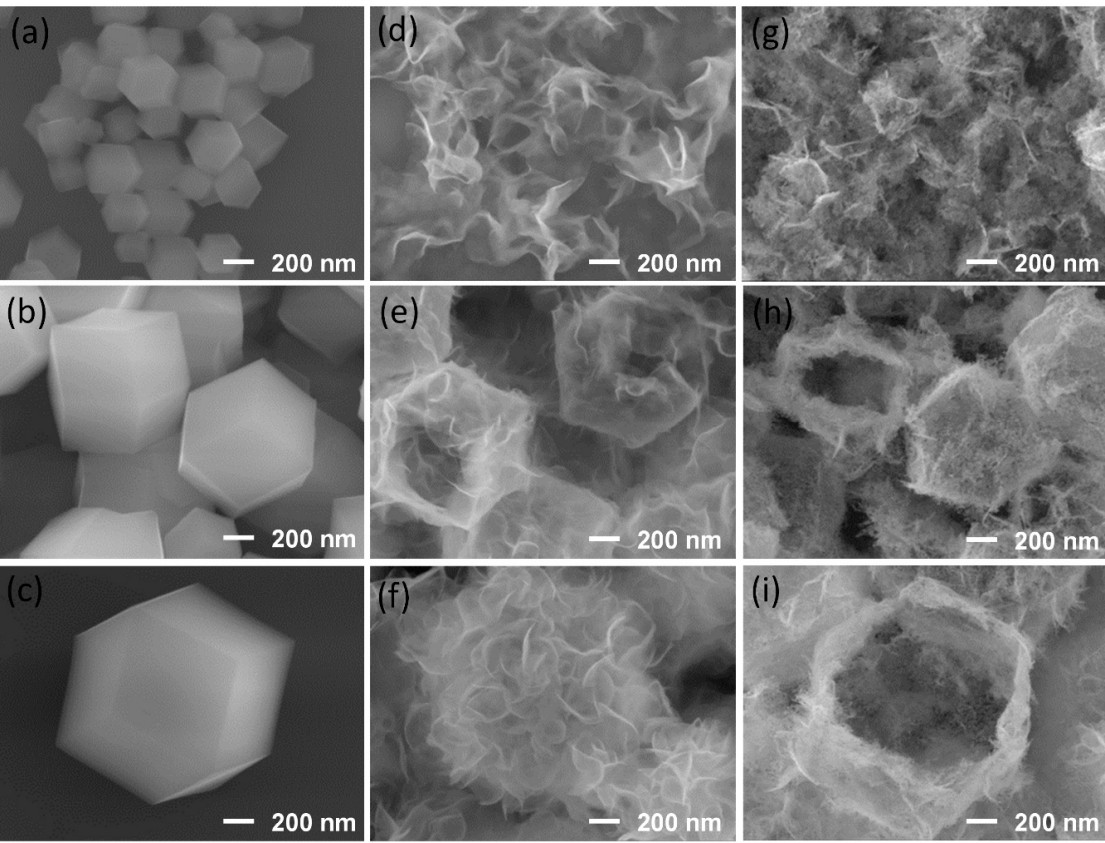

**Figure 2.** Scanning electron microscope (SEM) images of dodecahedral diamond zeolitic imidazolate framework-67 (ZIF-67): (**a**) ZIF-67-1, (**b**) ZIF-67-2, (**c**) ZIF-67-3; hollow Co-layered double hydroxide (Co-LDH): (**d**) Co-LDH-1, (**e**) Co-LDH-2, (**f**) Co-LDH-3; hollow $Co_3O_4$: (**g**) $Co_3O_4$-1, (**h**) $Co_3O_4$-2, (**i**) $Co_3O_4$-3.

The CV measurements in a potential range of 0–0.65 V (vs. Ag/AgCl) were first characterized to reveal the supercapacitors behavior of the $Co_3O_4$ electrodes. Figure 3a presents CV curves of the $Co_3O_4$ with different sizes at a scan rate of 5 mV/s. As expected, all samples deliver a distinct pair of peaks induced by a faradaic redox reaction, which clearly reveals the typical pseudocapacitive characteristics [23,24]. In particular, the integrated area under CV curve of $Co_3O_4$-2 electrode is the largest among three $Co_3O_4$ electrodes, demonstrating that the $Co_3O_4$-2 possesses the largest specific capacitance [34]. The CV curves of the $Co_3O_4$-2 electrode measured at typical scan rates from 5 to 100 mV/s are displayed in Figure 3b. The following classical and desirable pseudocapacitive phenomena were observed: (1) the CV curves are highly symmetrical; (2) the peak currents remarkably increase along with the increase of scan rates; (3) the peak position also changes accompanied by the increase of scan rates. The peaks generated in the anodic sweep shifted to a higher potential value, while the peaks which originated from the cathodic sweep moved towards a lower potential value; (4) no

significant distortion existed in a comparison of CV curves obtained at 5 and 100 mV/s. Notably, the obvious increase of peak currents and excellent symmetries in CV curves as well as negligible distortion indicate the rapid and well reversible redox process [30], further possibly resulting in an outstanding rate capability.

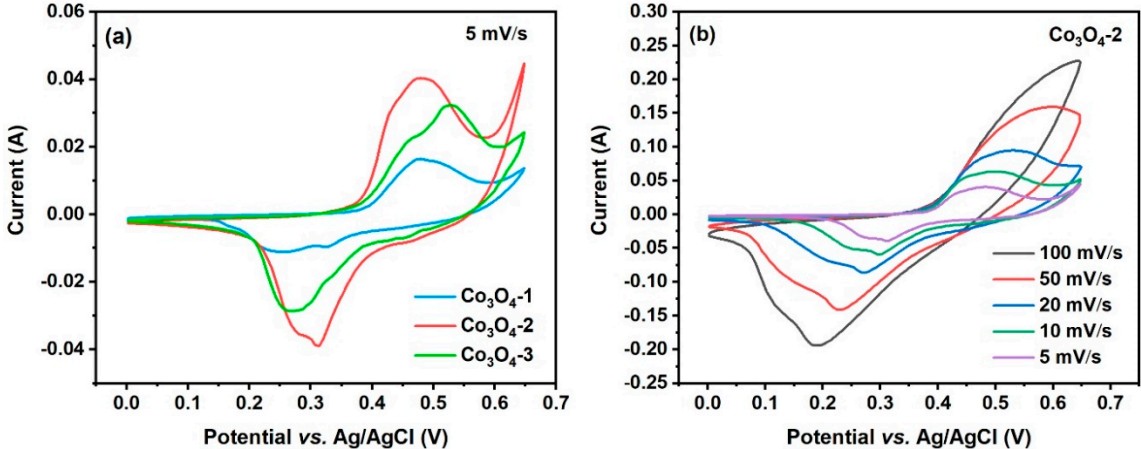

**Figure 3.** (**a**) Cyclic voltammetry (CV) curves of $Co_3O_4$-1, $Co_3O_4$-2 and $Co_3O_4$-3 electrodes at scan rate of 5 mV/s; (**b**) CV curves of $Co_3O_4$-2 electrode at typical scan rates ranging from 5 to 100 mV/s.

GCD tests between potential window of 0–0.65 V (vs. Ag/AgCl) were conducted to further investigate the capacitive properties. The discharge curves of $Co_3O_4$-1, $Co_3O_4$-2 and $Co_3O_4$-3 electrodes at representative current densities of 1, 2, 5, 10 and 20 A/g are presented in Figure 4a–c, respectively. It is intuitive that all discharge curves deliver evident plateaus coinciding with the redox reaction, which also confirms the pseudocapacitive behavior of each $Co_3O_4$ [25,30]. In addition, the GCD performances comparison at current densities of 1 A/g shown in Figure 4d also confirms the $Co_3O_4$-2 sample offers the largest specific capacitance [29], which is consistent with the aforementioned CV result (Figure 3a).

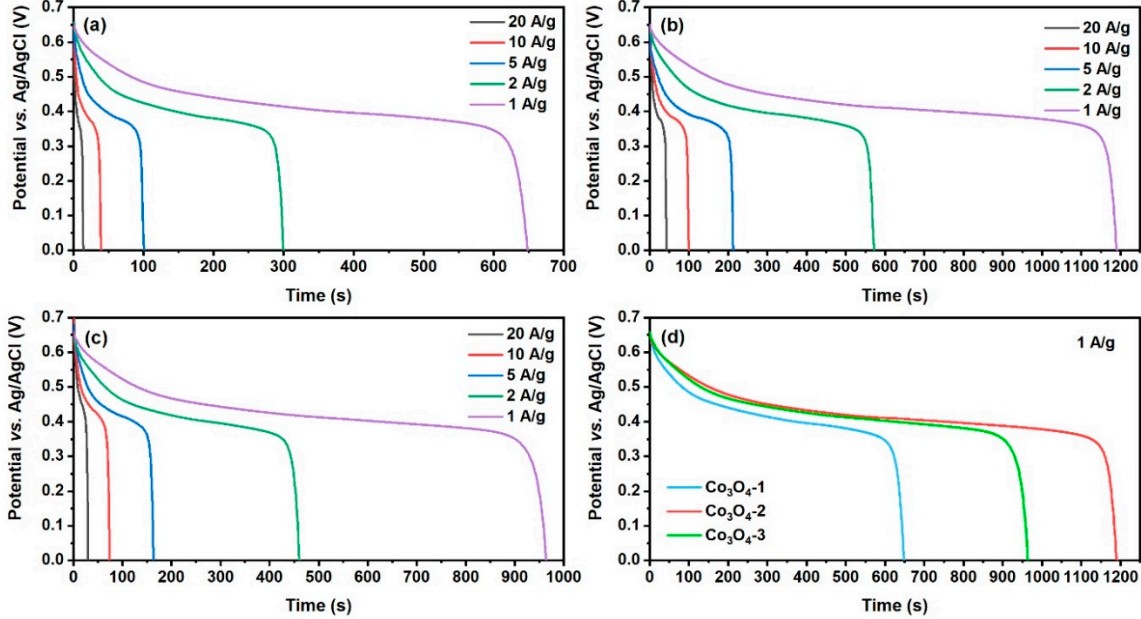

**Figure 4.** Discharge curves at representative current densities ranging from 1 to 20 A/g: (**a**) $Co_3O_4$-1, (**b**) $Co_3O_4$-2, (**c**) $Co_3O_4$-3; (**d**) discharge curves of $Co_3O_4$-1, $Co_3O_4$-2 and $Co_3O_4$-3 electrodes at current density of 1 A/g.

The specific capacitances based on GCD measurements are calculated according to Equation (1) for all $Co_3O_4$ electrodes, and the results are plotted in Figure 5. The $Co_3O_4$-2 electrode exhibits the specific capacitances of 1832.7, 1760.9, 1640.4, 1526.6 and 1324.5 F/g at 1, 2, 5, 10 and 20 A/g, respectively. Moreover, Figure 5 demonstrates that the boost in current densities leads to a decrease in specific capacitances. Unfortunately, the similar fading tendencies of $Co_3O_4$-1, $Co_3O_4$-2 and $Co_3O_4$-3 are inevitable owing to the correlation between current density and electrolytic ions diffusion (OH– in this study) in electrode material ($Co_3O_4$ in this study). Namely, compared with the diffusion at high current density, the OH– ions have relatively sufficient time to transfer at a low current density. The phenomenon gives rise to inner active sites failing to be comprehensively devoted to the redox reaction at high current density. In contrast, both the inner and outer surface can be involved at low current density, thoroughly making a redox reaction proceed and thus contributing to a large specific capacitance [35]. Nevertheless, the $Co_3O_4$-2 electrode can maintain capacitance of 72.3% from 1 to 20 A/g (43.5% of $Co_3O_4$-1, 62.6% of $Co_3O_4$-3). The slight decay provides the $Co_3O_4$-2 bright application with the high power required for energy storage devices. The relieved fading in specific capacitances can be explained by the unique morphology of synthesized $Co_3O_4$. During the redox process, the hollow structure can serve as an "OH– ions buffer reservoir" [35] to effectively minimize the influence of limited ions diffusion at a high current density on rate performance. In short, the impressive performances in specific capacitance and rate capability make the as-synthesized $Co_3O_4$-2 a promising electrode material in supercapacitors application.

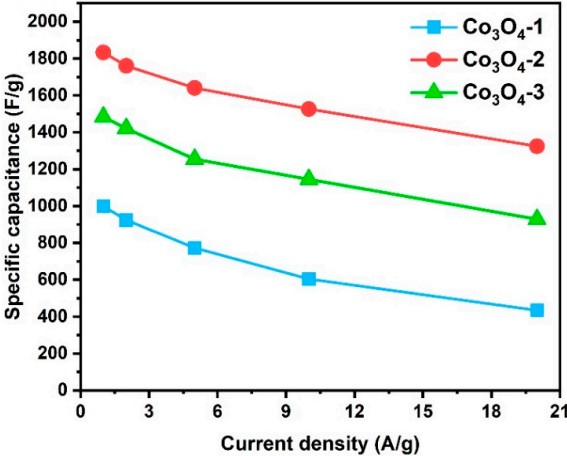

**Figure 5.** Specific capacitances of $Co_3O_4$-1, $Co_3O_4$-2 and $Co_3O_4$-3 electrodes as a function of current densities.

Figure 6a illustrates the Nyquist plots of different $Co_3O_4$ electrodes from frequency range of 100 mHz to 100 kHz. The plots of $Co_3O_4$-1, $Co_3O_4$-2 and $Co_3O_4$-3 are similar, which consists of a semicircle and a straight line. The intersection value of plot and real axis reflects the intrinsic resistance and interface resistance of electrolyte and active material (denoted as $R_s$) [35]. The $R_s$ of $Co_3O_4$-1, $Co_3O_4$-2 and $Co_3O_4$-3 are 0.55, 0.52 and 0.53 Ω, respectively (tiny difference). The semicircle acquired in high frequency range represents the double-layer capacitance ($C_{dl}$) and the charge-transfer resistance ($R_{ct}$, corresponds to the semicircle diameter). The semicircle also expresses that the $C_{dl}$ and $R_{ct}$ are connected in parallel [36]. Apparently, the $R_{ct}$ value of $Co_3O_4$-2 is much smaller than $Co_3O_4$-1 and $Co_3O_4$-3 (Figure 6b). In low frequency region, the approximate line in Nyquist plot is related to the interfacial diffusive process and determined by pseudocapacitance ($C_{ps}$) and Warburg element ($W$). The equivalent circuit is shown in Figure 6c. Benefited from the hollow structure constructed by numerous nanoparticles, the slops for all electrodes are larger than 45°, which presents an efficient electrolyte diffusion. Furthermore, a quasi-vertical line displayed by the plot of $Co_3O_4$-2 electrode implies a scarcely limited ions diffusion and the behavior tends to an ideal capacitor [23,36]. The

merits of $Co_3O_4$-2 electrode proposed in EIS results well coincide with the superior electrochemical performance of $Co_3O_4$-2 electrode in CV and GCD assessments.

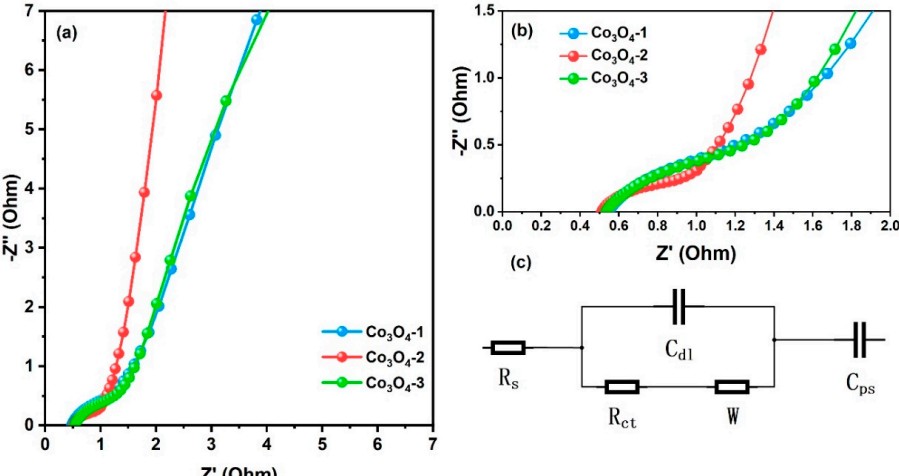

**Figure 6.** (**a**) Electrochemical impedance spectroscopy (EIS) plots of $Co_3O_4$-1, $Co_3O_4$-2 and $Co_3O_4$-3 electrodes; (**b**) enlarged view of high frequency region; (**c**) equivalent circuit involves intrinsic resistance and interface resistance ($R_s$), double-layer capacitance ($C_{dl}$), charge-transfer resistance ($R_{ct}$), Warburg element ($W$) and pseudocapacitance ($C_{ps}$).

The $Co_3O_4$-2 was employed as the optimized electrode through CV, GCD and EIS results for long term cycling investigation, as recorded in Figure 7. The evaluation was carried out by using a repetitive GCD process at 20 A/g. The $Co_3O_4$-2 electrode retains 85.9% of its initial specific capacitance even after undergoing 5000 cycles. The insets display the GCD curves of the $Co_3O_4$-2 electrode at first and last for 10 cycles. The coulombic efficiency ($\eta$) is also an important criterion for evaluating the durability and can be calculated based on Equation (2) [37]:

$$\eta = \frac{t_d}{t_c},$$

(2)

where $t_d$ (s) is the discharge time and $t_c$ (s) is the charge time. All deduced coulombic efficiency values reach over 95%. The excellent electrochemical stability of $Co_3O_4$-2 electrode is significant for possible commercial application.

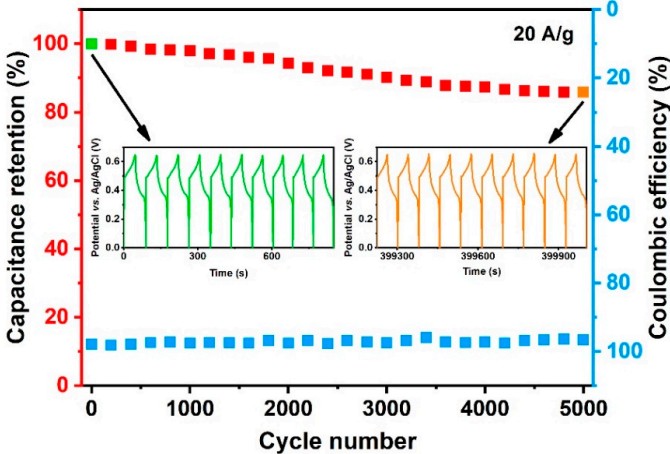

**Figure 7.** Cycling performance and coulombic efficiency of $Co_3O_4$-2 electrode. The insets are GCD curves at the first and last 10 cycles.

## 4. Discussion

In this study, the different electrochemical performances of $Co_3O_4$-1, $Co_3O_4$-2 and $Co_3O_4$-3 are determined by their own structural features. For $Co_3O_4$-1, mass collapsed structures accompanied the loss of opening pores, which were caused by ultrathin shells, while severely reducing the available surface area, as observed from a SEM image (Figure 2g). In contrast, during the fabricated process of $Co_3O_4$-2 and $Co_3O_4$-3, morphologies were well inherited not only in synthesis of Co-LDH from a ZIF-67 precursor but also in conversion of Co-LDH to $Co_3O_4$ (negligible aggregation). The hollow nature as well as the highly porous architecture (the shell is assembled by numerous nanoparticles) provide a large electrochemical active area with increased active sites for redox reactions. In addition, the unique morphology also significantly facilitates electron transportation and ion diffusion and permeation in electroactive material. The relatively lower electrochemical performance of $Co_3O_4$-3 electrode is possibly due to the decreased accessibility of electrolyte ions into material produced by the large size and thick shell [38–40]. Table 1 compares this work with other $Co_3O_4$ based electrodes reported in previous literature. It clearly demonstrates that the specific capacitance of hollow $Co_3O_4$ nanoboxes electrode in this research is competitive, and even higher than some $Co_3O_4$ hybrid materials. More importantly, the $Co_3O_4$ electrode reaches a specific capacitance of 1324.5 F/g at a high current density of 20 A/g. The value at 20 A/g is superior to some $Co_3O_4$ electrodes at 1 A/g. Except for the two criterions, a slight capacitance decay of 14.1% after 5000 cycles is also worth noting.

**Table 1.** Specific capacitances at low current density and high current density (rate performance) of $Co_3O_4$ based electrode materials.

| Material | Specific Capacitance (F/g)/ Low Current Density (A/g) | Specific Capacitance (F/g)/ High Current Density (A/g) | Ref. |
|---|---|---|---|
| hollow $Co_3O_4$ 3D-nanonet | 739/1 | 533/15 | [35] |
| hollow $Co_3O_4$ spheres | 474.8/1 | 377.4/10 | [37] |
| hollow $Co_3O_4$ spheres | 460/4 | 401/20 | [41] |
| hollow $Co_3O_4$ corals | 527/1 | 412/10 | [42] |
| hollow $Co_3O_4$ dodecahedron | 1100/1.25 | 437/12.5 | [43] |
| hollow $Co_3O_4$ nanowires | 599/2 | 439 /40 | [44] |
| $Co_3O_4$ nanowires | 977/2 | 484/10 | [45] |
| $Co_3O_4$ nanosheets | 1121/1 | 873/25 | [46] |
| $Co_3O_4$ nanoflakes | 448/0.5 | 421/10 | [47] |
| $Co_3O_4$ nanobooks | 590/0.5 | 421/8 | [48] |
| $Co_3O_4$ nanoplates | 355.6/0.4 | 230/4 | [49] |
| $Co_3O_4$ nanofibers | 340/1 | 296/10 | [50] |
| carbon incorporated $Co_3O_4$ | 978.9/0.5 | 303.3/15 | [51] |
| Mn doped $Co_3O_4$ | 773/1 | 485/16 | [52] |
| $Co_3O_4$/$CoMoO_4$ | 1902 /1 | 1200/10 | [53] |
| $Co_3O_4$/NiO | 1236.7/1 | 836.7/20 | [54] |
| $Co_3O_4$/$Ni(OH)_2$ | 1306.3/1.2 | 600 /12.1 | [55] |
| graphene/$Co_3O_4$ | 1765 /1 | 1266 /20 | [56] |
| rGO/$Co_3O_4$ | 546/0.5 | 496/5 | [57] |
| NiO/$Co_3O_4$/$MnO_2$ | 1055.3/0.2 | 727.4/4 | [58] |
| hollow $Co_3O_4$ nanoboxes | 1832.7/1 | 1324.5/20 | Ours |

## 5. Conclusions

In summary, hollow $Co_3O_4$ nanoboxes were synthesized successfully by using a facile method, in which the overall sizes of the $Co_3O_4$ were effectively controlled and as a consequence resulted in different supercapacitor performances. An overwhelming specific capacitance of 1832.7 F/g at 1 A/g and a comparable cycling stability of 85.9% after 5000 cycles are achieved in optimal $Co_3O_4$ electrode. In particular, the $Co_3O_4$ electrode still retains 72.3% of the specific capacitance, even at 20 A/g. The outstanding performances qualify the hollow $Co_3O_4$ nanoboxes an attractive electrode material in research field and commercial application of supercapacitors. Furthermore, the morphologic features

and tunable sizes make the applications of as-prepared $Co_3O_4$ promisingly extend to other aspects, especially in highly accessible surface area required fields, such as catalysts and gas sensors.

**Author Contributions:** Conceptualization, X.C.; Methodology, P.O. and X.C.; Investigation, X.F., P.O. and X.C.; Data Curation, X.F.; Formal Analysis, X.F.; Validation, X.F., P.O. and X.C.; Writing—Original Draft Preparation, X.F.; Writing—Review & Editing, P.O. and X.C.; Supervision, P.O. and X.C.; Project Administration, P.O. and X.C. All authors have read and agreed to the published version of the manuscript.

**Funding:** This research was mainly funded by Research Council of Norway (RCN), grant number 221860/F60.

**Acknowledgments:** X.F. is funded by China Scholarship Council (CSC), grant number 201506930018. The authors also thank the Norwegian Micro- and Nano-Fabrication Facility, NorFab, project number 245963 and Pai Lu for SEM characterization. Critical comments by Einar Halvorsen and Yongjiao Sun are acknowledged.

**Conflicts of Interest:** The authors declare no conflict of interest.

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
