# Peer review of "Tunable Synthesis of Hollow Co3O4 Nanoboxes and Their Application in Supercapacitors"

_applsci, doi:10.3390/app10041208_

Round 1

Reviewer 1 Report

Some remarks for the following manuscript:

-authors should add information about ZIF framework in the abstract,

-what was the magnification of sem images (materials characterization),

-add xrd measurement parameters, kind of lamp, (materials characterization),

-xrd patterns show strong background with very low intensity of Co3O4 signals correspond to relatively amorphous structure. Signals 400, 511, are hardly any visible, try to improve the pattern quality,

-give the name of abbreviation, line 88, NF,

-why the potential windowwas limited to 0.65V?,

-Authors suggest that Nyquist plots are similar, in fact Co3O4 is different, smaller loop, more steep high freq region, this should be defined.

Author Response

Authors should add information about ZIF framework in the abstract.

Response: The information about ZIF framework as well as relevant references was added to the abstract of the manuscript to make the background sufficient.

What was the magnification of SEM images (materials characterization).

Response: The magnification of SEM images is 30000.

Add XRD measurement parameters, kind of lamp, (materials characterization).

Response: The XRD patterns were performed on an EQUINOX 1000 with Cu-Kα1 radiation   (λ = 1.5406 Å) and the scanning speed was 6°/min. All parameters were added to the “Material Characterizations” section of the manuscript. 

XRD patterns show strong background with very low intensity of Co3O4 signals correspond to relatively amorphous structure. Signals 400, 511, are hardly any visible, try to improve the pattern quality.

Response: We smoothed XRD patterns of all samples to further improve the quality. Compared with original figure, the processed Co3O4 patterns are easily distinguished. In addition, we want to emphasize that in general XRD analysis, three indexed diffraction peaks are enough to confirm the final product.  

Figure 1. X-ray powder diffraction (XRD) patterns of Co3O4-1, Co3O4-2 and      Co3O4-3.

Give the name of abbreviation, line 88, NF.

Response: NF is the abbreviation of nickel foam. The NF was defined in parentheses the first time nickel foam appeared in line 91.

Why the potential window was limited to 0.65V?

Response: The potential window should ensures stable supercapacitors performance of electroactive material. In other words, no irreversible electrochemical reaction exist in suitable window. For alkaline electrolyte (KOH in this study), the positive voltage should be limited to 0.65 V to avoid possible oxygen evolution reaction.

Authors suggest that Nyquist plots are similar, in fact Co3O4 is different, smaller loop, more steep high freq region, this should be defined.

Response: The similarity we suggested means every Nyquist plot consists a semicircle and a straight line. The differences of Nyquist plots mainly focus on diameters of the semicircles and the slops of the lines. The diameter represents the charge-transfer resistance (Rct) and the slop is related to the electrolyte diffusion process. In this study, Co3O4-2 electrode delivers smallest diameter and largest slop, which is determined by its desirable morphological and structural features and coincides with its superior supercapacitors performance. The relationship between the structure and property was also discussed in detail in section 4 of the manuscript.  

Reviewer 2 Report

Regarding discussion of Fig. 2, specify that the values are for "particle size" Please increase the size of the scale bar marker in Fig. 2, to at least 200 nm, the current marker dimensions is quite small, even compared to the features in the micrographs Change use of "intuitionistic" to "intuitive" or similar word (e.g., logical) Include the descriptions of the terms in the equivalent circuit in the figure caption and explain physical meaning of "semicircle diameter/region"

Author Response

Regarding discussion of Fig. 2, specify that the values are for "particle size" Please increase the size of the scale bar marker in Fig. 2, to at least 200 nm, the current marker dimensions is quite small, even compared to the features in the micrographs.

Response: In discussion of Figure 2, the values of 0.39 ± 0.05, 1.98 ± 0.18 and 3.28 ± 0.37 μm are the average diameters of ZIF-67-1, ZIF-67-2 and ZIF-67-3, respectively. In discussion of Figure 1, the values of 23.6, 20.0 and 24.3 nm are the crystallite sizes of Co3O4-1, Co3O4-2 and Co3O4-3, respectively. The size of the scale bar marker was increased to 200 nm.

Figure 2. Scanning electron microscope (SEM) images of dodecahedral diamond zeolitic imidazolate framework-67 (ZIF-67): (a) ZIF-67-1, (b) ZIF-67-2, (c) ZIF-67-3; hollow Co-layered double hydroxide (Co-LDH): (d) Co-LDH-1, (e) Co-LDH-2, (f) Co-LDH-3; hollow Co3O4: (g) Co3O4-1, (h) Co3O4-2, (i) Co3O4-3.

Change use of "intuitionistic" to "intuitive" or similar word (e.g., logical) Include the descriptions of the terms in the equivalent circuit in the figure caption and explain physical meaning of "semicircle diameter/region".

Response: The “intuitionistic” was replaced by “intuitive”. The equivalent circuit terms descriptions were also added to the figure caption. In EIS study, the Nyquist plot typically consists a semicircle in high frequency range and a straight line in low frequency region. The diameter of the semicircle corresponds to the charge-transfer resistance (Rct) which is an element in equivalent circuit.